# The Evolution of the WUSCHEL-Related Homeobox Gene Family in *Dendrobium* Species and Its Role in Sex Organ Development in *D. chrysotoxum*

**DOI:** 10.3390/ijms25105352

**Published:** 2024-05-14

**Authors:** Xiaoting Luo, Qinyao Zheng, Xin He, Xuewei Zhao, Mengmeng Zhang, Ye Huang, Bangping Cai, Zhongjian Liu

**Affiliations:** Key Laboratory of National Forestry and Grassland Administration for Orchid Conservation and Utilization at College of Landscape Architecture, Fujian Agriculture and Forestry University, Fuzhou 350002, China; 1221775039@fafu.edu.cn (X.L.); qinyaozheng@fafu.edu.cn (Q.Z.); 5220422102@fafu.edu.cn (X.H.); zxw6681@163.com (X.Z.); 1220428020@fafu.edu.cn (M.Z.); 5221726097@fafu.edu.cn (Y.H.)

**Keywords:** WOX gene family, *Dendrobium*, floral development, expression pattern

## Abstract

The WUSCHEL-related homeobox (WOX) transcription factor plays a vital role in stem cell maintenance and organ morphogenesis, which are essential processes for plant growth and development. *Dendrobium chrysotoxum*, *D. huoshanense*, and *D. nobile* are valued for their ornamental and medicinal properties. However, the specific functions of the WOX gene family in *Dendrobium* species are not well understood. In our study, a total of 30 WOX genes were present in the genomes of the three *Dendrobium* species (nine DchWOXs, 11 DhuWOXs, and ten DnoWOXs). These 30 WOXs were clustered into ancient clades, intermediate clades, and WUS/modern clades. All 30 WOXs contained a conserved homeodomain, and the conserved motifs and gene structures were similar among WOXs belonging to the same branch. *D. chrysotoxum* and *D. huoshanense* had one pair of fragment duplication genes and one pair of tandem duplication genes, respectively; *D. nobile* had two pairs of fragment duplication genes. The *cis*-acting regulatory elements (CREs) in the WOX promoter region were mainly enriched in the light response, stress response, and plant growth and development regulation. The expression pattern and RT-qPCR analysis revealed that the WOXs were involved in regulating the floral organ development of *D. chrysotoxum*. Among them, the high expression of *DchWOX3* suggests that it might be involved in controlling lip development, whereas *DchWOX5* might be involved in controlling ovary development. In conclusion, this work lays the groundwork for an in-depth investigation into the functions of WOX genes and their regulatory role in *Dendrobium* species’ floral organ development.

## 1. Introduction

The homeobox transcription factors (HB TFs) are key regulators of plant and animal cell fates and differentiation, and homeobox genes were first discovered in Drosophila [1,2]. Meanwhile, more homeobox members continue to be found in other eukaryotes. The WUSCHEL (WUS) gene is the prototypic member of the plant-specific WUS homeobox (WOX) protein family, one of several HB TF families [3]. A total of 14 homologs of *AtWUS* were searched in the *Arabidopsis* genome, and these genes were named WOXs [4]. The WOX TFs contain a short stretch of amino acids that folds into a DNA-binding domain (called a homeodomain), which forms helix–loop–helix–turn–helix structures in space [5].

In plants, the WOX genes are extensively distributed. According to the evolutionary origins among genes, the members of the WOX gene family in plants can be clustered into three clades: the ancient clade, the intermediate clade, and the WUS/modern clade. All plant species (from algae to angiosperms) contain varying amounts of WOX genes belonging to the ancient clade; the intermediate clade is found in pteridophytes, gymnosperms, and angiosperms; and the WUS/modern clade is exclusively found in angiosperms [6]. In the WOX gene family of *Arabidopsis*, there are three members (*AtWOX10*, *AtWOX13–14*) in the ancient clade, four members (*AtWOX8–9*, *AtWOX11–12*) in the intermediate clade, and eight members (*AtWUS*, *AtWOX1–7*) in the WUS/modern clade.

The WOX gene family is involved in plant growth and development, as well as in the stress response. WOX family members belonging to different clades fulfill different biological functions in the development of plant flowers, floral meristems, roots, and other organs. The WOX genes of the ancient clade participate in the regulation of plant roots and flower development. *AtWOX13* is expressed in floral meristem tissues, inflorescences, and young flower buds and is particularly highly expressed in developing carpels. *WOX13* promotes replum development by negatively regulating the *JAG/FIL* genes [7]. *AtWOX14* is found only in Brassicaceae, where it is expressed early in lateral root formation and specific to the development of anthers [8]. The intermediate clade mainly affects embryo patterning and root organogenesis. *WOX8* and *WOX9* are homologous genes that play vital roles in embryo and inflorescence development and are species-specific in their functions [2,9,10,11]. The genes *WOX11* and *WOX12*, which are homologous, participate in the process of de novo root organogenesis in *Arabidopsis* [12]. The WUS clade mainly affects the development of the floral meristem and leaf and stem cell maintenance. For example, *Arabidopsis* WUS genes can maintain stem cell homeostasis at all developmental stages in the shoot apical meristem (SAM) [13,14]. Meanwhile, WUS genes are also able to act as activators to regulate the size of the floral meristem tissue [15]. *WOX1* and *WOX3* redundantly regulate abaxial–adaxial growth in the leaf and floral meristem [16,17]. *AtWOX2* is required to initiate the embryogenic shoot meristem stem cell program in *Arabidopsis* [18]. *WOX5* is critical for stem cell maintenance in the root apical meristem (RAM) [19,20]. In addition, the WOX gene family plays an important role in the response to environmental stresses, such as salt, cold, and drought. For example, *GhWOX4* positively regulates drought tolerance in cotton; *PagWOX11/12a* positively regulates the salt tolerance of poplar [21,22].

The WOX genes act as transcription factors to activate or repress the expression of other genes on the one hand, as described above for the role played by WOXs in plants. On the other hand, the upstream part of the WOX coding region contains abundant CREs to receive the action of other regulatory factors. For example, maize *ZMSP10/14/26* regulates the expression of the *ZmWOX3A* gene in coat precursor cells by directly binding to its promoter [23]. In summary, the combination of cis- and trans-acting factors exerts a regulatory effect on gene expression, while playing an indispensable role in plant growth, development, and evolution [24].

Orchidaceae, one of the largest angiosperm groups, contains over 750 genera and 28,000 species [25,26]. It is widely distributed, with the exception of the North and South Poles and extremely arid desert areas, and has the greatest distribution in the tropics. Orchids are highly evolved taxa within angiosperms and are one of the most studied taxa in biological research [27]. *Dendrobium* is the second-largest genus in the orchid family and is a typical epiphyte [28]. Most *Dendrobium* species have valuable medicinal stems, while their flowers and leaves have excellent ornamental value. In recent years, the completion of the whole-genome sequencing of *D. catenatum* [29], *D. chrysotoxum* [30], *D. huoshanense* [31], and *D. nobile* [32], etc., has provided valuable information revealing the genetic and molecular mechanisms of the formation of important traits in *Dendrobium*. The regulatory function of the WOX genes in model plants such as *Arabidopsis* has been relatively comprehensively researched. However, there is little knowledge about how the WOX genes affect the growth and development of *Dendrobium* species.

In our study, we identified the WOX gene family in three *Dendrobium* species (*D. chrysotoxum*, *D. huoshanense*, and *D. nobile*), and systematically analyzed their basic traits, including their chromosomal localization, phylogenetics, motif compositions, gene structures, collinearity, and CREs. Meanwhile, the expression pattern of WOXs in the *D. chrysotoxum* flower parts was analyzed. This project aimed to preliminarily elucidate the evolutionary and potential biological roles of the WOX gene family in *Dendrobium* species, and to provide new insights into the study of the molecular regulatory mechanisms of the WOX genes in *D. chrysotoxum* flower development.

## 2. Results

### 2.1. Identification and Physicochemical Properties of the WOX Gene Family

The WOX genes in three *Dendrobium* species were screened by BLAST and HMMER. The result showed that ten, nine, and 11 WOXs were identified in the genomes of *D. chrysotoxum*, *D. huoshanense*, and *D. nobile*, respectively. According to the order distribution of the chromosomes, these WOXs were named *DchWOX1–10*, *DhuWOX1–9*, and *DnoWOX1–11*.

To characterize the WOX genes of the *Dendrobium* species in more detail, we predicted the physicochemical properties of 30 WOX proteins using ExPASy. The results are as follows (Table 1). The number of amino acids (AA) varied from 110 aa (*DchWOX1*) to 328 aa (*DchWOX4*), and the molecular weight (Mw) ranged from 12.84 kDa (*DchWOX1*) to 35.30 kDa (*DnoWOX4*). Among the 30 WOXs, 12 were basic proteins with an isoelectric point (pI) higher than 8.00; the remaining 17, with a pI ranging from 5.26 (*DhuWOX8*) to 7.79 (*DnoWOX4*), were neutral or weakly acidic proteins. Additionally, the grand average of hydrophilic (GRAVY) values of all WOX proteins were less than zero, suggesting their strong hydrophilicity. The instability indexes (II) of all WOX members exceeded 40, implying that these proteins are unstable [33]. All WOX proteins were found to be in the nucleus according to subcellular location predictions, indicating that they might also function there, like most TFs.

### 2.2. Chromosomal Localization of WOXs

As illustrated in Figure 1A, ten DchWOXs were unevenly present on six chromosomes of *D. chrysotoxum* (Chr01, 06, 08, 12, 15, and 19) (Figure 1A). Nine DhuWOXs were distributed on six chromosomes, Chr4, 6, 7, 11, 13, and 19, of *D. huoshanense* (Figure 1B). The results of the chromosome mapping for *D. nobile* showed that 11 DnoWOXs were distributed across seven chromosomes (Figure 1C). In addition, we observed a pair of tandem repeat genes in *D. chrysotoxum* (*DchWOX2* and *DchWOX3*) and *D. huoshanense* (*DhuWOX6* and *DhuWOX7*).

### 2.3. Phylogenetic Analysis of WOXs

We created a phylogenetic tree of the WOX genes to analyze the evolution of the WOX genes in the *Dendrobium* species (Figure 2). The evolutionary tree included 30 WOXs from three *Dendrobium* species, 15 AtWOXs from *A. thaliana*, and 13 OsWOXs from *O. sativa*. All WOX protein sequences have been collected with Appendix A. According to the classification of the WOXs’ evolutionary relationships in *A. thaliana*, the 30 WOXs in the *Dendrobium* species can be similarly clustered into the ancient clade (six WOX genes), the intermediate clade (ten WOX genes), and the WUS/modern clade (14 WOX genes). The WUS/modern clade has the largest number of WOX genes, while the ancient clade has the fewest.

### 2.4. Gene Structure and Conserved Motifs of WOXs

The conserved motifs of the 30 WOXs in the three *Dendrobium* species were evaluated through the online prediction website MEME (Figure 3B). The results demonstrated that, whereas the motif structures varied by clade, WOXs within the same clade had comparable motif structures. Ten conserved motifs were detected in the 30 WOXs. Appendix A has listed all motif sequences. All WOXs contain motif 1 and motif 3 simultaneously; motif 6, motif 7, and motif 10 are exclusive to the WUS/modern clade; and motif 9 is found only in the intermediate clade. The distinct roles of various WOXs may be conferred by the particular distributions of various structures.

We visualized the number and distribution of the WOXs’ introns and exons to further reveal the gene structures of the WOXs in the three *Dendrobium* species (Figure 3C). Most *Dendrobium* WOXs contain 1–2 introns. Notably, three introns were detected in *DnoWOX4* and four introns were detected in *DchWOX9*, while *DchWOX1* and *DhuWOX5* had no introns. The gene structures of WOX members belonging to the same clade are similar. In particular, in the ancient clade, the phylogenetic tree divides six genes into two structurally similar subclades. *DhuWOX8*, *DnoWOX5*, and *DchWOX6* are clustered as subclades with two introns, while *DchWOX10*, *DhuWOX1*, and *DnoWOX11* are clustered as a subclade with one intron.

Multiple sequence pairs of the 30 WOXs showed that all WOXs contained a helix–turn–helix–loop–helix region unique to the homeodomain (Figure 4A). Twelve WOXs in the WUS/modern clade contain the WUS-box (TL-LFP-) (Figure 4B). 

### 2.5. Synteny Analysis and Ka/Ks Value of WOX Gene Family

The *D. chrysotoxum* genome contains a pair of segmental duplication genes, *DchWOX4* and *DchWOX8* on Chr06 and Chr15 (Figure 5A). Similarly, the *D. huoshanense* genome contains one pair of fragment duplication genes, *DhuWOX2* and *DhuWOX5* on Chr6 and Chr11 (Figure 5B). Two pairs of segmental duplicates were found in the *D. nobile* genome, *DnoWOX2* and *DnoWOX9* on CM039718.1 and CM039732.1, and *DnoWOX3* and *DnoWOX8* on CM039723.1 and CM039732.1, respectively (Figure 5C). Furthermore, the *Ka*/*Ks* ratios of these four gene pairs were all less than 0.5, ranging from 0.13 to 0.2 (Table 2).

### 2.6. Cis-Acting Elements Analysis of WOXs

We extracted 2000 bp upstream of the CDS of the 30 WOX genes to identify the CREs to predict the potential regulatory functions of the WOX genes in the *Dendrobium* species. In total, 569 CREs belonging to 35 types and 17 response functions were found in the three *Dendrobium* species (Figure 6 and Appendix A).

We classified the retrieved CREs into four categories: growth and development elements, phytohormone responsiveness, stress repressiveness, and light responsiveness. The growth and development element category includes endosperm expression, circadian control, and meristem expression. Interestingly, among them, the frequency of meristem expression is the largest. Five types of CRE exist within the category of phytohormone responsiveness. This includes abscisic acid (ABA), methyl jasmonate (MeJA), auxin, gibberellin, and salicylic acid responsiveness. The stress repressiveness category had four types of CRE, including defense and stress responsiveness, anaerobic induction, drought, and low-temperature stress, with anaerobic induction being the most frequent. In addition, light responsiveness accounts for almost half of all CREs (269/569), and there is a large frequency of light responsiveness in each WOX gene.

As shown in Figure 6C, *DchWOX9* in *D. chrysotoxum* has the largest number (33 CREs) of elements, *DhuWOX2* and *DhuWOX4* in *D. huoshanense* have the most (25 CREs), and *DnoWOX6* in *D. nobile* has the most (34 CREs).

### 2.7. Expression Pattern Analysis of WOX Gene Family in D. chrysotoxum

We performed expression analyses based on transcriptome data from different flower parts in the three developmental periods of *D. chrysotoxum* (Figure 7). In the transcriptome heatmap, *DchWOX6* and *DchWOX10* of the ancient clade were expressed at significantly higher levels in S1. However, *DchWOX6* was similarly expressed in all five floral parts, whereas *DchWOX10* exhibited high expression only in the gynostemium. Of the three members of the intermediate clade, *DchWOX5* had higher expression in the ovary of S1, *DchWOX8* in the sepal of S1, and *DchWOX4* had lower expression in S1 than S2 and S3. Among the five WOXs of the WUS/modern clade, *DchWOX2* and *DchWOX3* displayed similar expression levels throughout flower development, and they had higher expression amounts in the lip of S1; *DchWOX7* had higher expression in the ovary of S1; *DchWOX1* was highly expressed in the ovary of S2; and *DchWOX9* gynostemium expression was highest in S1.

### 2.8. RT-qPCR Analysis of WOX Genes in D. chrysotoxum

We selected *DchWOX3*, *DchWOX5,* and *DchWOX10* from different clades for RT-qPCR experiments to further elucidate the expression patterns of the WOXs during the development of different flower parts in *D. chrysotoxum* (Figure 8). As shown, the *DchWOX3* RT-qPCR results are in general agreement with the transcriptome data, i.e., *DchWOX3* showed very low expression in other parts of the flower, while it was significantly expressed in the S1 lip, and its expression was gradually downregulated during flower development (Figure 8A). *DchWOX5* was consistently expressed in the ovary during the three periods, suggesting that *DchWOX5* is involved in regulating ovary development (Figure 8B). The transcriptome expression heatmap showed that *DchWOX10* was significantly expressed during S1 in the gynostemium. However, the RT-qPCR results indicated a trend of increasing followed by decreasing expression of *DchWOX10* (Figure 8C). These differences may have resulted from imperfect correlations between the samples used for transcriptome sequencing and the samples used for RT-qPCR.

## 3. Discussion

A class of TFs unique to plants, the WOX family is involved in critical regulatory functions in important developmental programs like organ morphogenesis and stem cell maintenance [3,34]. In our study, 30 WOXs were identified from the genomes of three species: *D. chrysotoxum* contained nine DchWOXs, *D. huoshanense* contained 11 DhuWOXs, and *D. nobile* contained 10 DnoWOXs. The three *Dendrobium* species had similar numbers of WOXs to *D. catenatum* (14) [35], *Phalaenopsis equestris* (10) [36], *A. thaliana* (15), *O. sativa* (13), *Sorghum bicolor* (11) [37], *Vitis vinifera* (12) [38], tobacco (9) [39], and *Triticum aestivum* (14) [40], but differed from those of dicotyledonous plants such as *Brassica napus* (58) [41], soybean (33) [42], and *Gossypium hirsutum* (40) [43]. Variations in the size of a species’ genome or processes like gene and genome replication may be the cause of the variations in the number of WOXs between species [44].

All three *Dendrobium* species in this study had undergone at least two whole-genome duplication (WGD) events [30,31,32]. According to the chromosome distribution map (Figure 1), both *D. chrysotoxum* and *D. huoshanense* harbored a single pair of tandem repeat genes. The synteny analysis showed that both *D. chrysotoxum* and *D. huoshanense* had a single pair of genes with segmental duplication, and *D. nobile* had two pairs of genes with segmental duplication (Figure 5). It is probably because of these duplication events that the WOX genes differed in number and distribution among the three species. In addition, the *Ka/Ks* ratios of the four WOX gene pairs detected in this study were all less than one, revealing that these WOXs underwent strong purifying selection during evolution (Table 2) [45]. This enables them to remain highly conserved in evolving *Dendrobium* species, maintaining the specific biological functions of WOX proteins [46].

The phylogenetic analysis of the 30 WOXs from the *Dendrobium* species compared with those from *Arabidopsis* and *O. sativa* showed that the distribution of the WOXs in *Dendrobium* species is conserved (Figure 2). Like most plants, such as *O. sativa*, *Picea abies*, and *Eriobotrya japonica*, the WUS/modern clade had the highest number of WOX genes and the ancient clade had the lowest number of WOX genes among the three *Dendrobium* species [36,47,48]. The loss of certain WOX genes occurred in the three *Dendrobium* species, except for *WOX1/6/7/8/14*, which is unique to dicotyledons. For example, *D. chrysotoxum* and *D. huoshanense* both lost *WOX4*, and only *D. nobile* retained the homologous gene for *AtWOX4* (*DnoWOX6*). Strikingly, *DchWOX1* and *DnoWOX1* were well clustered into a subclade with *AtWUS* and *OsWOX1* (*AtWUS* homologous gene). *AtWUS* was shown to be the prototype of the *Arabidopsis* WOX gene family, so we speculate that *DchWOX1* is the prototype of the WOXs of *D. chrysotoxum*, and *DnoWOX1* is the prototype of the WOXs of *D. nobile* [3]. However, similar to *D. catenatum*, the prototype gene was absent in *D. huoshanense* [35]. We hypothesize that, during evolution, there may have been functional redundancy among WOX family members to compensate for the functions performed by the missing genes, or some species-specific WOXs may have arisen [3,39,49].

Supported by the conserved motifs and intron patterns, the highly conserved gene structure guarantees the conserved function of each clade or subclade. WOX genes in the same subfamily tend to have similar numbers of introns and exons, and they also share similarities in gene structure (Figure 3) [50,51]. The gene of the ancient clade has a more conserved gene structure than the other two clades’ genes, consistent with the WOX genes of *Arabidopsis*, *Poplar*, and *Sorghum* [37]. The conserved ancient clade is present in all plants, and we hypothesize that strict conservation ensures that these WOX proteins perform indispensable functions in plant evolution [6]. The concatenated motif 3 and motif 1 (Figure 3A) correspond to the homeodomain sequence shown in Figure 4 and are present in all 30 WOX proteins. The homeodomain exhibits a helix–turn–helix–loop–helix structure, which ensures that it can differentiate between sequence-specific targets with precise spatiotemporal organization (Figure 4A) [52]. Similar to most plants, such as *Arabidopsis*, rice, and maize, only WUS/modern clade members contained the WUS-box (TL-LFP-) (Figure 4B) [4,39]. In summary, the sequence and structure conservation of the WOX gene family members maintains their functional integrity across species.

Transcriptional regulation occurs mainly through the promoter and its associated CREs to activate or repress gene expression [53]. The WOX gene family is extensively involved in regulating the development of various plant organs and contributes to abiotic stress and phytohormone signaling. The promoter regions of these 30 WOXs were rich in light-responsive elements, suggesting that WOXs play an essential role in regulating the light response (Figure 6) [54]. The meristem expression is the most frequent of the growth and developmental components. *DchWOX4*, *DhuWOX2*, and *DnoWOX3* had two, three, and three meristem expression elements, respectively (Figure 6B), and these three genes shared a branch with *AtWOX11* and *AtWOX12*. Since *AtWOX11* and *AtWOX12* participate in new root organ development in *Arabidopsis* [12], *DchWOX4*, *DhuWOX2*, and *DnoWOX3* are speculated to regulate the differentiation of roots in the three *Dendrobium* species, respectively. The stress-repressive CREs in the promoter region of the WOXs mainly include anaerobic induction, drought inducibility, and low-temperature responsiveness, which implies that the WOX genes are essential for plants to respond to abiotic stresses. This has been verified in *Arabidopsis* and *O. sativa*; for example, the rab21 promoter drives *OsWOX13* overexpression in *O. sativa*, thereby improving its drought tolerance [55]. Elements associated with the plant hormone response in the promoter region of the WOXs are ABA, IAA, SA, GA, and MeJA. Many studies have revealed that the WOX is affected by IAA, ABA, and GA during plant growth and development [22,56]. The MeJA response element is the most abundant phytohormone response element. It is involved in plant defense responses and also regulates plant growth and development. [57,58]. To summarize, the WOX gene family in *Dendrobium* species has a vital function in plant growth, development, and the stress response by mediating phytohormone regulation.

The transcriptome analysis of *Arabidopsis*, *O. sativa*, *Fragaria vesca*, and *Nelumbo nucifera* indicated that *NnWOX14* was significantly expressed in the carpel of *N. nucifera*; *FvWOX9* and *FvWOX9a* in *F. vesca* showed significant expression in the process of development; and WOX family members are expressed in the flowers of both *Arabidopsis* and *O. sativa* [37,59,60]. All of these observations suggest the significance of the WOX gene in the formation of floral organs in plants. Therefore, we combined transcriptomic data from the flowers of *D. chrysotoxum* with RT-qPCR experiments to identify the important regulatory role of the WOX genes in *D. chrysotoxum* flower development (Figure 7 and Figure 8). According to the results of *DchWOX3* being significantly expressed in the S1 lip (Figure 8A), along with the development process of gradually reducing the amount of its expression, we speculate that *DchWOX3* may participate in regulating lip growth. It has been found that *PeWOX9A*, *PeWOX9B*, and *DcWOX9* are highly expressed in the gynoecium in *D. catenatum* and *P. equestris*, respectively, and the overexpression of *DcWOX9* in *Arabidopsis* resulted in staminate and pistil sterility [36]. In our study, *DchWOX5* was significantly expressed in the ovary during the three periods (Figure 8B). *PeWOX9A*, *PeWOX9B*, *DcWOX9,* and *DchWOX5* both share a clade with *AtWOX9*, indicating their specific roles in regulating gynoecium and ovary development, which need to be further verified.

## 4. Materials and Methods

### 4.1. Data Sources

The genome files of *D. chrysotoxum* (accession number: PRJNA664445) and *D. nobile* (accession number: PRJNA725550) were retrieved from the NCBI (https://www.ncbi.nlm.nih.gov/, accessed on 25 September 2023). The genome file of *D. huoshanense* (accession number: CNA0014590) was retrieved from the China Nucleotide Sequence Archive (CNSA, https://ftp.cngb.org/, accessed on 25 September 2023). A total of 15 *A. thaliana* WOX protein sequences were retrieved from TAIR (http://www.arabidopsis.org, accessed on 25 September 2023). The 13 WOX protein sequences of *O. sativa* were retrieved from PlantTFDB (http://planttfdb.gao-lab.org/, accessed on 25 September 2023).

### 4.2. Identification and Physicochemical Properties of WOXs 

Candidate WOX genes were searched in the genomes of three *Dendrobium* species in the two-way BLAST tool of the TBtools v2.003 software, using the 15 WOXs of *A. thaliana* as probes, respectively [61,62]. Meanwhile, using the Simple HMM Search tool of TBtools, the Hidden Markov Model (HMM) file of the homeodomain (PF00046) from the Pfam database (http://pfam.xfam.org/search, accessed on 27 September 2023) was utilized to further identify WOX family members in the three *Dendrobium* species. Candidate WOXs identified by BLAST and HMM were uploaded to NCBI CD-Search (https://www.ncbi.nlm.nih.gov/Structure/cdd/wrpsb.cgi, accessed on 27 September 2023) for structural analysis, and only genes with conserved typical homeodomains of WOXs were retained.

All finalized sequences of the 30 WOX proteins were uploaded to the online software ExPASy (https://www.expasy.org/, accessed on 27 September 2023) for physicochemical property analysis, to obtain the amino acids (aa), molecular weight (MW), isoelectric point (pI), instability index (II), aliphatic index (AI), and grand average of hydropathicity (GRAVY) of all WOX proteins [63]. Then, the subcellular localization prediction of the WOX family members was performed by the online program Cell-PLoc 2.0 (http://www.csbio.sjtu.edu.cn/bioinf/Cell-PLoc-2/, accessed on 27 September 2023).

### 4.3. Chromosomal Localization

Based on the annotation data of the three *Dendrobium* genomes, chromosomal localization maps of the WOX genes were produced using Gene Location Visualize from the GTF/GFF program of TBtools. 

### 4.4. Phylogenetic Analysis of WOX Gene Family

The protein sequences of 15 AtWOXs, 13 OsWOXs, ten DchWOXs, nine DhuWOXs, and 11 DnoWOXs were uploaded into the MEGA11 software, and then these 58 WOX protein sequences were used to achieve sequence alignment using the Clustal W function (default parameters), and the phylogenetic trees of the five species were constructed using the maximum likelihood (1000 bootstrap replication) [64,65]. The editing and beautification of the phylogenetic tree was performed by Evolview 3.0. (http://www.evolgenius.info/evolview/#/treeview, accessed on 8 October 2023) [66].

### 4.5. Protein Conservative Domain and Gene Structure Analysis

The prediction of conserved structural domains for the 30 WOXs in *Dendrobium* was accomplished by utilizing the CDD program from NCBI (https://www.ncbi.nlm.nih.gov/cdd, accessed on 10 October 2023). The identification of conserved motifs for the 30 WOXs was performed by the MEME online program (https://meme-suite.org/meme/tools/meme, accessed on 10 October 2023) [67]. Gene Structure View in TBtools was employed to map the phylogenetic trees, conserved motifs, and gene structures in combination. The WOX protein sequence alignment was performed by Clustal W of MEGA 11 and then beautified by jalview (Version: 2.11.3.2). 

### 4.6. Synteny Analysis of WOX Gene Family

The identification of intra-species duplicate genes in the three *Dendrobium* species was performed using the One Step MCScanx function of TBtools [68]. In Advance Circos of TBtools, the duplication patterns of the three *Dendrobium* species were visualized. Then, the calculation of the *Ka*, *Ks*, and *Ka/Ks* values for the gene pairs was accomplished by the Simple *Ka/Ks* Calculator in TBtools.

### 4.7. Cis-Acting Regulatory Element Analysis

First, Gtf/Gff3 Sequence Extract and Fasta Extract of TBtools were used to extract 2000 bp upstream of the 30 WOX genes. Second, to complete the prediction of the CREs, the acquired sequences were submitted to the online website PlantCARE (http://bioinformatics.psb.ugent.be/webtools/plantcare/html/, accessed on 13 October 2023). Finally, the distribution of the acquired CREs was visualized using the Basic Biosequence View module of TBtools, while the categories and number of CREs were counted and plotted in Excel 2016 [69].

### 4.8. Expression Pattern and RT-qPCR Analysis

*D. chrysotoxum* plant materials were taken from the Forest Orchid Garden of Fujian Agriculture and Forestry University for transcriptome sequencing and RT-qPCR, including five flower parts (sepal, petal, lip, ovary, and gynostemium) in three periods (unpigmented bud stage, pigmented bud stage, and early flowering stage).

The transcriptome sequencing and library construction of the five flower parts from the three periods of *D. chrysotoxum* development were performed by BGI Genomics Co., Ltd. (Shenzhen, China). RESM v1.2.8 was used for transcript quantification and to calculate the FPKM value for each sample. Based on the FPKM value, heatmaps of gene expression are created in the HeatMap program of TBtools.

Further validation of the expression patterns of the three WOX genes was achieved by RT-qPCR experiments. The FastPure Plant Total RNA Isolation Kit (for polysaccharide- and polyphenol-rich tissues) (Vazyme Biotech Co., Ltd., Nanjing, China) was used to extract total RNA from *D. chrysotoxum* samples. The Hifair^®^ AdvanceFast One-Step RT-gDNA Digestion SuperMix for qPCR (YEASEN, Shanghai, China) was used to generate the cDNA for the quantitative PCR. Based on the transcription data, *DchActin* (*Maker75111*) was selected as the reference gene. The WOX gene sequences were submitted to the Primer Premier 5 software to design specific PCR primers (Appendix A). The TSINGKE ArtiCanATM SYBR qPCR Mix was used for the RT-qPCR analysis on the Bio-Rad/CFX Connect Real-Time PCR Detection System (Bio-Rad Laboratories, Hercules, CA, USA). Three biological replicates were carried out for all experiments. Finally, the relative expression of the three WOX genes was calculated using the 2^−∆∆CT^ method with S1 Se as the reference. The data were visualized using GraphPad Prism 7.0.

## 5. Conclusions

In this study, 10, 11, and 9 WOX genes were identified in the genomes of *D. chrysotoxum*, *D. huoshanense*, and *D. nobile*, respectively, and chromosomal localization, phylogeny, gene structure, and motif composition analyses were performed. In addition, based on the transcriptome and RT-qPCR experiments, we analyzed the expression patterns of the DchWOXs in five floral parts of *D. chrysotoxum* at three developmental periods. In conclusion, our results provide useful information for the in-depth exploration of the biological roles of the WOX gene family, as well as floral developmental studies in *Dendrobium* species.

## Figures and Tables

**Figure 1 ijms-25-05352-f001:**
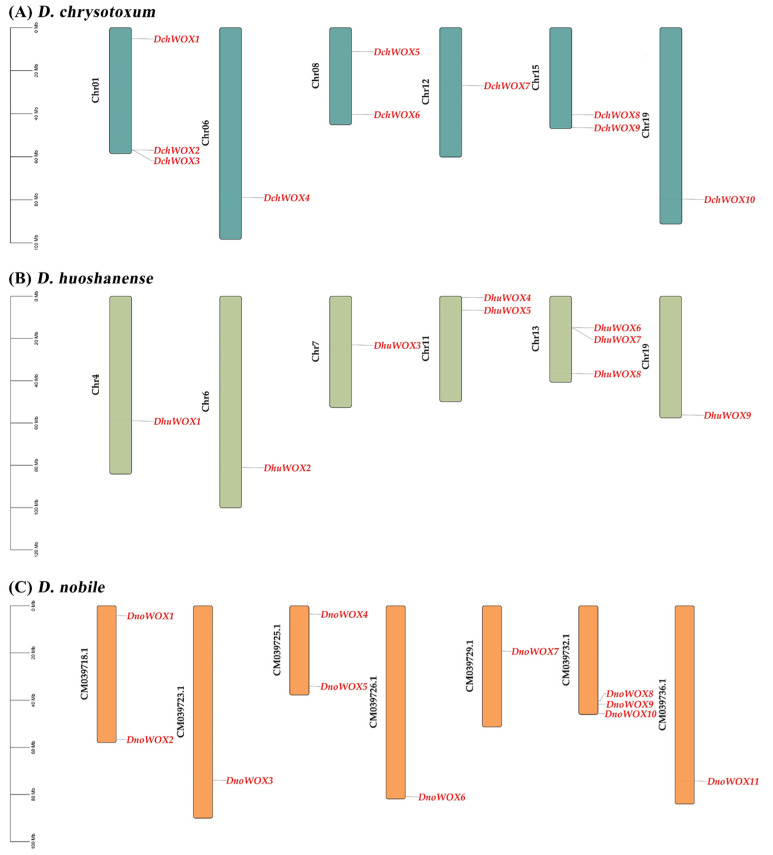
Chromosome distribution of WOX genes. (**A**) *D. chrysotoxum*. (**B**) *D. huoshanense*. (**C**) *D. nobile*. Black labels are chromosome names and red labels are gene names.

**Figure 2 ijms-25-05352-f002:**
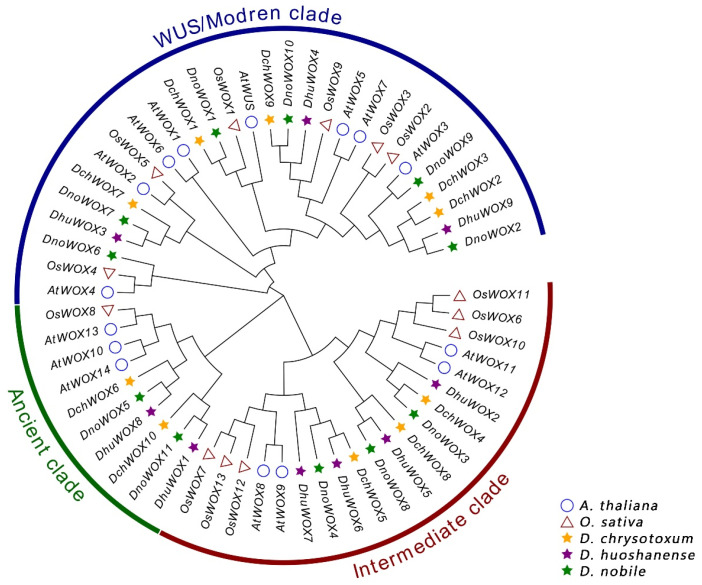
Phylogenetic tree of WOXs in *A. thaliana*, *O. sativa*, and three *Dendrobium* species.

**Figure 3 ijms-25-05352-f003:**
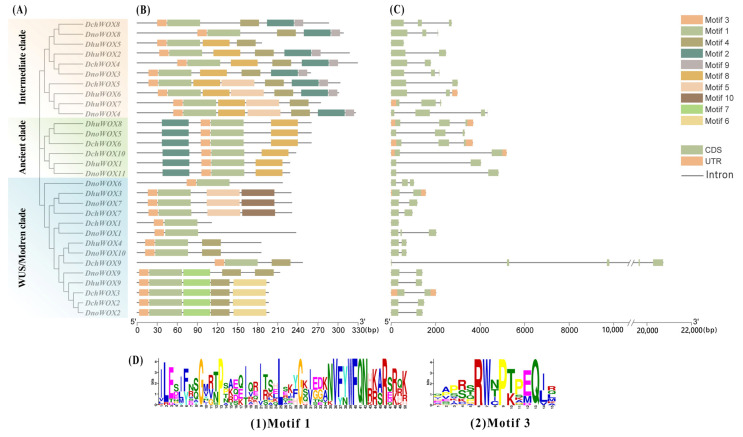
Phylogenetic tree, motifs, and structures of WOXs in three *Dendrobium* species. (**A**) Phylogenetic tree of 30 WOXs. (**B**) Conserved motifs of 30 WOX proteins. (**C**) Intron and exon structures of 30 WOX genes. (**D**) Sequence logos of motif 1 and motif 3.

**Figure 4 ijms-25-05352-f004:**
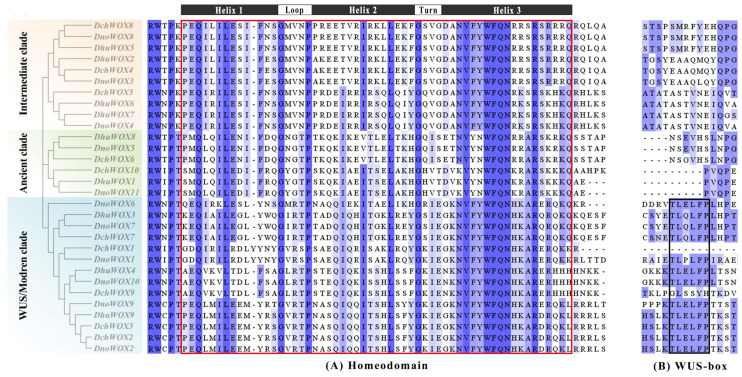
Multiple sequence alignment results of the WOX gene family in three *Dendrobium* species. (**A**) Homeodomain. (**B**) WUS-box. The red box indicates the homeodomain and the black box indicates the WUS-box domain.

**Figure 5 ijms-25-05352-f005:**
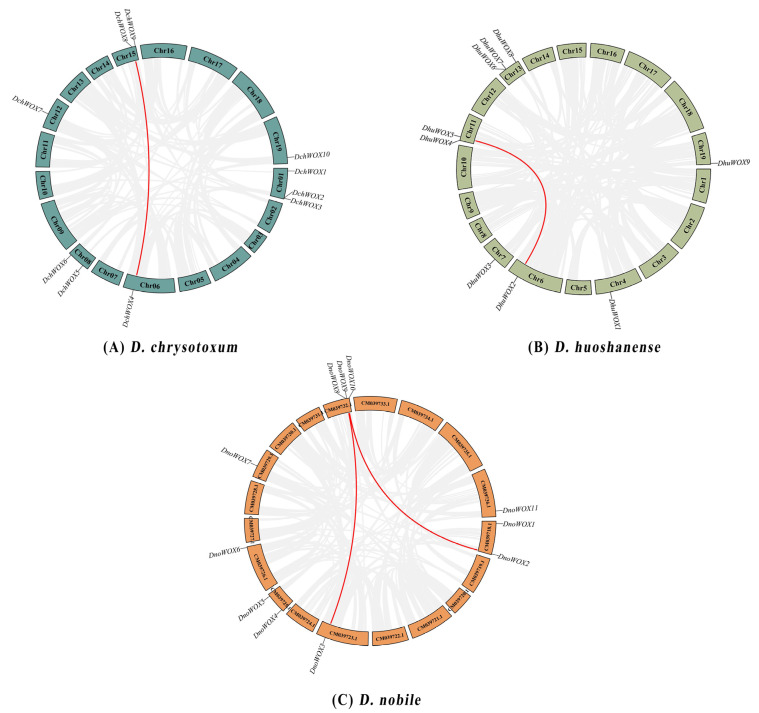
Synteny analysis of the WOX gene family. (**A**) *D. chrysotoxum*. (**B**) *D. huoshanense.* (**C**) *D. nobile*. The red lines refer to segmental duplicate gene pairs.

**Figure 6 ijms-25-05352-f006:**
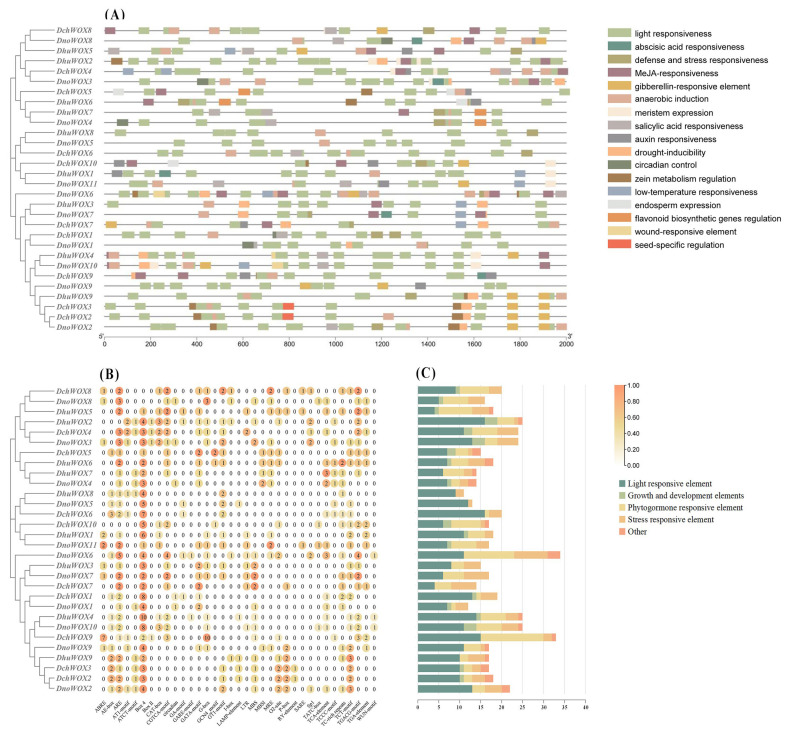
The CREs in the promoter regions of 30 WOX genes. (**A**) Distribution of the WOX CREs; (**B**) the number of CREs; (**C**) statistics on the number of different categories of CREs. The types and numbers of CREs are listed in Appendix A.

**Figure 7 ijms-25-05352-f007:**
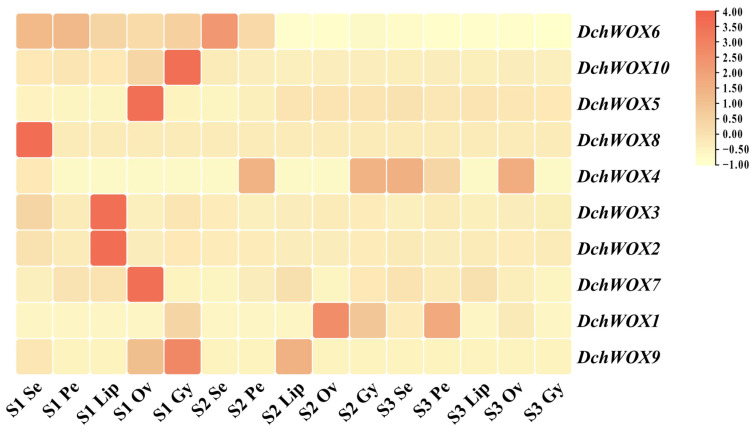
The expression levels of ten DchWOXs in *D. chrysotoxum* at five floral parts and three developmental periods. S1: unpigmented bud stage; S2: pigmented bud stage; S3: early flowering stage; Se: sepal; Pe: petal; Lip: lip; Ov: ovary; Gy: gynostemium. Appendix A lists the FPKM values for the WOXs in *D. chrysotoxum*.

**Figure 8 ijms-25-05352-f008:**
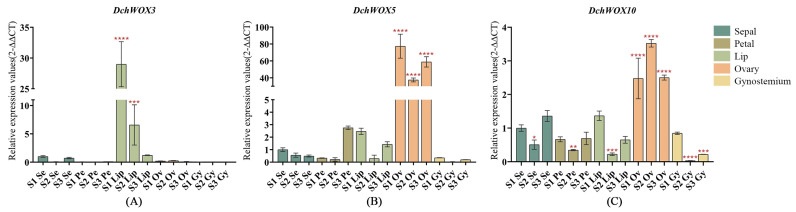
RT-qPCR verified the expression of DchWOXs in the developing flower organs of *D. chrysotoxum*. (**A**) *DchWOX3*; (**B**) *DchWOX5*; (**C**) *DchWOX10*. The Y-axis represents the relative expression values (2^−∆∆CT^). The error bars indicate three RT-qPCR biological replicates. *p*-values in the significance test are indicated by red asterisks (* *p* < 0.05, ** *p* < 0.01, *** *p* < 0.001, **** *p* < 0.0001). Appendix A shows the primer sequences of the DchWOXs and reference gene.

**Table 1 ijms-25-05352-t001:** Characteristics of the WOXs from three *Dendrobium* species.

Gene Nane	Number of Amino Acids (aa)	Molecular Weight (kDa)	Theoretical PI	Instability Index	Aliphatic Index	Grand Average of Hydropathicity	Subcellular Localization
*DchWOX6*	259	29.82	5.34	66.87	58.76	−0.887	Nucleus
*DchWOX8*	285	31.33	7.68	65.78	70.07	−0.395	Nucleus
*DchWOX4*	328	35.07	8.37	70.06	79.73	−0.065	Nucleus
*DchWOX3*	195	22.15	9.11	68.49	53.03	−0.772	Nucleus
*DchWOX7*	230	25.85	6.32	77.88	61.09	−0.693	Nucleus
*DchWOX2*	195	22.14	9.11	69.91	53.03	−0.772	Nucleus
*DchWOX10*	236	27.19	6.27	67.38	64.87	−0.834	Nucleus
*DchWOX5*	302	33.16	6.24	55.35	91.06	−0.098	Nucleus
*DchWOX9*	246	28.41	9.47	74.03	77.72	−0.706	Nucleus
*DchWOX1*	110	12.84	9.69	69.36	69.36	−1.195	Nucleus
*DhuWOX4*	184	21.06	9.08	60.70	73.10	−0.684	Nucleus
*DhuWOX7*	273	29.67	9.48	70.79	71.14	−0.472	Nucleus
*DhuWOX1*	227	26.35	6.20	66.47	71.28	−0.717	Nucleus
*DhuWOX3*	229	25.90	7.00	75.10	60.09	−0.691	Nucleus
*DhuWOX6*	300	32.64	6.31	63.71	88.43	−0.170	Nucleus
*DhuWOX8*	259	29.91	5.26	65.97	60.27	−0.878	Nucleus
*DhuWOX5*	185	19.89	8.98	79.63	54.38	−0.556	Nucleus
*DhuWOX9*	196	22.14	9.11	65.29	53.78	−0.758	Nucleus
*DhuWOX2*	316	34.27	7.66	62.10	75.66	−0.110	Nucleus
*DnoWOX11*	227	26.31	6.20	66.47	68.72	−0.750	Nucleus
*DnoWOX7*	230	26.04	6.60	74.71	59.39	−0.694	Nucleus
*DnoWOX4*	325	35.30	7.79	61.37	76.86	−0.364	Nucleus
*DnoWOX6*	216	24.38	9.91	69.27	71.34	−0.748	Nucleus
*DnoWOX2*	196	22.16	9.11	69.22	53.78	−0.760	Nucleus
*DnoWOX8*	307	33.27	6.30	69.20	75.28	−0.173	Nucleus
*DnoWOX10*	184	21.15	9.39	61.85	74.67	−0.664	Nucleus
*DnoWOX5*	259	29.92	5.26	66.63	60.27	−0.876	Nucleus
*DnoWOX1*	236	26.49	5.75	55.23	55.08	−0.838	Nucleus
*DnoWOX3*	258	27.53	5.78	66.90	69.26	−0.258	Nucleus
*DnoWOX9*	212	24.24	6.79	78.91	48.77	−0.922	Nucleus

**Table 2 ijms-25-05352-t002:** The Ka/Ks of the WOX gene family in the three *Dendrobium* species.

Gene 1	Gene 2	*K*a	*K*s	*K*a/*K*s
*DchWOX4*	*DchWOX8*	0.332789	1.919410	0.173381
*DhuWOX5*	*DhuWOX2*	0.305003	2.326057	0.131124
*DnoWOX2*	*DnoWOX9*	0.226893	1.109762	0.204452
*DnoWOX3*	*DnoWOX8*	0.267376	1.697561	0.157506

## Data Availability

The genome sequences and annotation files of *D. chrysotoxum* (accession number: PRJNA664445) and *D. nobile* (accession number: PRJNA725550) were downloaded from the National Center for Biotechnology Information (NCBI, https://www.ncbi.nlm.nih.gov/, accessed on 25 September 2023) genome database. The genome file of *D. huoshanense* (accession number: CNA0014590) was downloaded from the China Nucleotide Sequence Archive (CNSA, https://ftp.cngb.org/, accessed on 25 September 2023).

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
