# Peer review of "The Evolution of the WUSCHEL-Related Homeobox Gene Family in Dendrobium Species and Its Role in Sex Organ Development in D. chrysotoxum"

_ijms, 2024, doi:10.3390/ijms25105352_

Round 1
Reviewer 1 Report
Comments and Suggestions for Authors
The manuscript titled with" Identification and Characterization of the WUSCHEL-related homeobox Gene Family in Dendrobium Species and Its Expression Analysis in D. chrysotoxum". The idea of the manuscript is very good and I impressed for the obtained results. But I have some comments which could be summarized in:
Title: should be change into; Evolution of WUSCHEL-related homeobox Gene Family in Dendrobium Species and Its role sex organs development in D. chrysotoxum".
Abstract: One line on the importance of studied plant D. chrysotoxum. In line 25, In summary should be substituted by" in collectively" .
Introduction:
Line 69 in the introduction should be the line 31, we always start with the importance of the studied materials. So, please rearrange the introduction.
Apragrap on cis-acting and trans acting genes and their role in plant development and evolution should be add to the introduction section.
Also, the expression of the family genes is conditional or not conditional.
The aim is not quite clear and I see that the authors hesitate to write sharp sentences about their aim.
Results:
It is written in good manner.
Discussion the length of the discussion is very good but the citation is so little. Please add more references and interpretation the work in good manner.
Conclusion
Authors should add an conclusion even the journal structure not permit with that.
Author Response
Response to Reviewer 1 Comments
Thank you very much for taking the time to review this manuscript. Please find the detailed responses below and the corresponding revisions in the re-submitted files.
Comment 1. Title should be change into "Evolution of WUSCHEL-related homeobox Gene Family in Dendrobium Species and Its role sex organs development in D. chrysotoxum".
Response: Thank you for your comments. We think this is a great suggestion and have changed the title.
Comment 2. In line 25, In summary should be substituted by" in collectively ".
Response: Thank you for your comments. We have changed "In summary" to "In collectively". [Please see Line 25 in Word's revision mode]
Comment 3. Line 69 in the introduction should be the line 31, we always start with the importance of the studied materials. So, please rearrange the introduction.
Response: Thank you for the suggestion. We have revised the introduction. In the third paragraph of the Introduction, we would like to elaborate that the WOX gene family has functions in regulating both plant growth and development and stress response. Considering that our study focuses on the role of WOX in plant growth and development, the content related to stress response was only briefly elaborated. We hope our revision is acceptable to you. [please see Line 50-51 and Line 72-73 in Word's revision mode]
Comment 4. Apragrap on cis-acting and trans acting genes and their role in plant development and evolution should be add to the introduction section.
Response: Thank you for your comments. We have added WOX genes as a cis and trans gene and their role in plant development and evolution related to the preface. [please see Line 74-81 in Word's revision mode]
Comment 5. The expression of the family genes is conditional or not conditional.
Response: Thank you for your comments. In our study, the expression analysis regarding the WOX gene family was unconditional. We did not perform any special treatment on the plant materials. The plant material for transcriptome and RT-qPCR experiments were flower parts of D. chrysotoxum under normal growth conditions, which were taken from three different developmental periods. It is not clear to me whether this suggestion of yours is correctly understood by me. If my understanding is wrong, I sincerely hope that you can explain this issue again.
Comment 6. The aim is not quite clear and I see that the authors hesitate to write sharp sentences about their aim.
Response: Thank you for your reminding. We have revised the sentence about the purpose of this paper, and hope that it will be beneficial to clearly express the purpose of our study. [please see Line 99-102 in Word's revision mode]
Comment 7. Discussion: the length of the discussion is very good but the citation is so little. Please add more references and interpretation the work in good manner.
Response: Thank you for your comments. We have added references in the discussion, such as. [45], [46], [49], [50], [51],and [54], in the hope of increasing the persuasiveness of the discussion section.
Comment 8. Authors should add an conclusion even the journal structure not permit with that.
Response: Thank you for your comments. We have added a conclusion section at the end of the manuscript to make the article more structured. [please see Line 431-439 in Word's revision mode]

Reviewer 2 Report
Comments and Suggestions for Authors
The Authors present a paper in which they report on their study of WOX genes in three Dendrobium species. The study concerns the identification and evaluation of the regulatory elements and organisation of these genes, and the analysis of their gene product, amino acid structure and function. That the article will provide a great deal of information is already clear from the abstract and the introduction in which the functions of the products of the known WOX genes are given.
At the end of the introduction the aim of the study is clear and introduces the very detailed results. As in all studies in which the analysis is mostly computer-based, so much data is obtained that the figures and tables are much needed to make the reading easy, making the information more comprehensible even to those less familiar with this type of study. The supplementary material also helps in the compression of the study.
The materials and methods are well described. The results are clear and well discussed, what is missing is at the end of the discussion one or two sentences of precise and representative conclusions (if not an actual paragraph of conclusions) as such a work deserves, preferably indicating possible further developments of the study.
some notes for the authors:
- attention to the spaces, which are sometimes missing before the brackets as on lines 35, 64, 196
- at line 94 I would indicate the three dendrobium species in which the 10, 9 and 11 WOX genes have been identified respectively
- at line 99 seems to be missing a word indicating that the content as number of amino acids varies in the different gene products of the WOX genes (The amino acid (AA) varied from…)
- in many figures the wording is really very small (e.g. in figures 1, 3 and 5)
Comments on the Quality of English Language
Minor editing of English language is necessary
Author Response
Response to Reviewer 2 Comments
Thank you very much for taking the time to review this manuscript. Please find the detailed responses below and the corresponding revisions in the re-submitted files.
Comment 1. The results are clear and well discussed, what is missing is at the end of the discussion one or two sentences of precise and representative conclusions (if not an actual paragraph of conclusions) as such a work deserves, preferably indicating possible further developments of the study.
Response: Thank you for your suggestion. We added a small modification to the end of the discussion. [please see Line 344-345 in Word's revision mode]. Also, taking into account the comments of other reviewers, we have added a conclusion section at the end of the manuscript to make the article more structured. [please see Line 431-439 in Word's revision mode]
Comment 2. Attention to the spaces, which are sometimes missing before the brackets as on lines 35, 64, 196.
Response: Thank you for the heads up. We apologize for our carelessness. We have added spaces and rechecked the whole article. [please see Line 35, 65, 212, 251, 285 in Word's revision mode]
Comment 3. At line 94 I would indicate the three dendrobium species in which the 10, 9 and 11 WOX genes have been identified respectively
Response: Thank you for your comments. We have made modifications to point out the three Dendrobium species for which 10,9 and 11 WOX genes were identified, respectively. [please see Line 105-107 in Word's revision mode]
Comment 4. At line 99 seems to be missing a word indicating that the content as number of amino acids varies in the different gene products of the WOX genes (The amino acid (AA) varied from…)
Response: Thank you for reminding me. We have filled in the missing words. [please see Line 112 in Word's revision mode]
Comment 5. In many figures the wording is really very small (e.g. in figures 1, 3 and 5)
Response: Thank you for reminding me. We made adjustments to the figures content, enlarged the figures, and adjusted the layout to hopefully present a clearer figure content. [please see figures 1, 3, 4 and 5 in Word's revision mode]

Reviewer 3 Report
Comments and Suggestions for Authors
Abstract:
The abstract provides a concise overview of the study, highlighting the identification and characterization of the WUSCHEL-related homeobox (WOX) gene family in Dendrobium species and its expression analysis in D. chrysotoxum. It effectively summarizes the key findings, including the total number of WOX genes identified, their distribution across different clades, conserved motifs and gene structures, as well as their expression patterns in D. chrysotoxum floral organs. However, it could be improved by providing more specific details regarding the methods used for gene identification and expression analysis.
Introduction:
The introduction comprehensively discusses the significance of the WUSCHEL-related homeobox (WOX) gene family in plant development and provides relevant background information on its evolutionary origins and functional diversity. It effectively sets the context for the study by highlighting the lack of understanding regarding the role of WOX genes in Dendrobium species. The references cited support the statements made, contributing to the credibility of the introduction. However, there are some minor grammatical errors and awkward phrasings that could be revised for clarity.
Results:
The results section presents detailed findings on the identification, phylogenetic analysis, gene structure, chromosomal localization, and expression patterns of WOX genes in Dendrobium species. The data are well-organized and supported by figures and tables, enhancing the clarity and understanding of the results. However, there are some inconsistencies in numbering and labeling of figures and tables that need to be addressed. Additionally, some figures lack detailed legends explaining the content, which could be confusing for readers.
Discussion:
The discussion section provides a thorough analysis and interpretation of the results, relating them to existing knowledge in the field. It effectively discusses the implications of the findings and their significance in advancing our understanding of WOX gene function in Dendrobium species. However, it would benefit from further exploration of the potential biological implications of the observed expression patterns and evolutionary conservation of WOX genes.
Materials and Methods:
The materials and methods section outlines the experimental procedures used for gene identification, phylogenetic analysis, and expression analysis. It provides sufficient detail for reproducibility, including data sources, software tools, and analytical methods. However, there are some typographical errors and inconsistencies in formatting that should be corrected.
Suggestions to Authors:
Clarity and Consistency: Ensure consistency in numbering and labeling of figures and tables throughout the manuscript. Provide detailed legends for all figures to enhance reader comprehension.
Grammar and Syntax: Review the manuscript for grammatical errors and phrasings, particularly in the introduction and discussion sections. Consider revising sentences for clarity and coherence.
Biological Implications: Expand the discussion to include a more in-depth analysis of the biological implications of the observed expression patterns and evolutionary conservation of WOX genes in Dendrobium species.
Proofreading: Consider having the manuscript proofread by a native English speaker or professional editor to identify and correct any grammatical errors or language inconsistencies.
Questions to Authors:
1. Can you provide more insight into the potential functional significance of the identified conserved motifs in WOX proteins?
2. How do you plan to validate the predicted roles of specific WOX genes in floral organ development through experimental approaches?
3. Could the observed differences in expression patterns between transcriptome data and RT-qPCR results be attributed to technical variations or biological factors?
Overall, the manuscript presents valuable insights into the identification and
characterization of the WOX gene family in Dendrobium species. Addressing the minor issues mentioned above would further enhance the clarity and quality of the manuscript.
Comments on the Quality of English LanguageRecommended for English editing
Author Response
Response to Reviewer 3 Comments
Thank you very much for taking the time to review this manuscript. Please find the detailed responses below and the corresponding revisions in the re-submitted files.
Comment 1. Clarity and Consistency: Ensure consistency in numbering and labeling of figures and tables throughout the manuscript. Provide detailed legends for all figures to enhance reader comprehension.
Response: Thanks for the reminder. We have checked the images and tables of the manuscript again, correcting inconsistencies in numbering and labeling. At the same time, all diagrams requiring legends were checked.
Comment 2. Grammar and Syntax: Review the manuscript for grammatical errors and phrasings, particularly in the introduction and discussion sections. Consider revising sentences for clarity and coherence.
Response: Thank you for your suggestions. We have corrected the incorrect grammar and wording in the manuscript. We have not listed the changes here, but have marked them in red in the revised manuscript.
Comment 3. Biological Implications: Expand the discussion to include a more in-depth analysis of the biological implications of the observed expression patterns and evolutionary conservation of WOX genes in Dendrobium species.
Response: Thank you for your suggestions. We have added to our discussion with a more in-depth analysis of the biological significance of WOX gene expression patterns and evolutionary conservation. We hope that our modifications will be recognized by you. [Please see Line 269-271, Line 277-280, and Line 285-288 in Word's revision mode]
Comment 4. Proofreading: Consider having the manuscript proofread by a native English speaker or professional editor to identify and correct any grammatical errors or language inconsistencies.
Response: Thank you for your suggestions. We have done our best to proofread the manuscript and correct any grammatical errors or language inconsistencies. We hope that our corrections are approved.
Comment 5. Can you provide more insight into the potential functional significance of the identified conserved motifs in WOX proteins?
Response: Thank you for your question. Two important conserved structural domains are present in the WOX gene family proteins, the homeodomain, which consists of motif 1 and motif 3, and the WUS-box, which is present only in the WUS clade. [Please refer to Figure 3 and Figure 4 in the manuscript.]
Plants have gradually diversified their organ morphology during the evolutionary process, and the different morphologies of plants are related to the activity of stem cells. Previous studies have shown that the WOX family functions in a wide range of developmental processes from stem cell maintenance to embryonic pattern formation, including branching, floral meristem formation, root formation, etc. The homeodomain is the main region where WOX family members carry out these functions, and the high degree of conservatism of the homeodomain maintains the key role of WOX family members in plant evolution.
WUS clade members are involved in regulating the development of many types of meristematic tissues. The WUS-box is a structural domain that is required for all activities of WUS clade members present only in seed plants and which may have a transcriptional repressive role. Previous studies have shown that the initial function of the WOX gene is transcriptional activation, while the repressive function was possessed by the WUS clade through the acquisition of the structural domain, WUS-box, during the evolutionary process.
However, homologous WOX genes diverge in different plant species with different biological functional differentiation, so we conducted preliminary studies on WOX in three Dendrobium species in the hope of providing reliable reference data for a comprehensive understanding of the potential functions and regulatory mechanisms of WOX during plant development.
Comment 6. How do you plan to validate the predicted roles of specific WOX genes in floral organ development through experimental approaches?
Response: Thank you for your question. We considered that our prediction can be further verified by heterologous overexpression experiments. That is, an overexpression vector for a specific WOX gene was constructed and transfected into A. thaliana to observe whether there is any abnormality in the development of floral organs in A. thaliana. Then, transcriptome analysis and RT-qPCR experiments were performed on the abnormally developed floral organs to further verify the potential biological role of this gene in flower development.
Comment 7. The Could the observed differences in expression patterns between transcriptome data and RT-qPCR results be attributed to technical variations or biological factors?
Response: Thank you for your question. We consider that this can be attributed to either technical differences or biological factors. In terms of technical differences, there are some differences between the two techniques themselves, RT-qPCR and transcriptome. It is also possible that the differences in results are due to the way individuals operate RT-qPCR. In terms of biological factors, it is difficult for us to guarantee that the experimental materials for RT-qPCR and transcriptome are in perfect agreement. In our study, the two samples of D. chrysotoxum used for RT-qPCR and transcriptome were taken from different plants, which may have caused some differences in the results.
